# Machine Learning for Precise Rice Variety Classification in Tropical Environments Using UAV-Based Multispectral Sensing

Arif K. Wijayanto [1,2,3,*] , Ahmad Junaedi [4], Azwar A. Sujaswara [5], Miftakhul B. R. Khamid [4,6],
Lilik B. Prasetyo [1] , Chiharu Hongo [7] and Hiroaki Kuze [7]

1   Department of Forest Resources Conservation and Ecotourism, Faculty of Forestry and Environment,
    IPB University, Bogor 16680, Indonesia; lbprastdp@apps.ipb.ac.id
2   Japan Society for Promotion of Science (JSPS) Ronpaku Fellow, Tokyo 102-0083, Japan
3   Environmental Research Center, IPB University, Bogor 16680, Indonesia
4   Department of Agronomy and Horticulture, Faculty of Agriculture, IPB University, Bogor 16680, Indonesia;
    junaedi_agr@apps.ipb.ac.id (A.J.); miftakhul.bakhrir@staff.unsika.ac.id (M.B.R.K.)
5   Graduate School of Agriculture, Kyoto University, Kyoto 606-8502, Japan; azwar_sujaswara@apps.ipb.ac.id
6   Program of Agrotechnology, Faculty of Agriculture, Universitas Singaperbangsa Karawang,
    Karawang 41361, Indonesia
7   Center for Environmental Remote Sensing (CEReS), Chiba University, Chiba 263-8522, Japan;
    hongo@faculty.chiba-u.jp (C.H.); hkuze@faculty.chiba-u.jp (H.K.)
*   Correspondence: akwijayanto@apps.ipb.ac.id; Tel.: +62-251-8621947

**Abstract:** An efficient assessment of rice varieties in tropical regions is crucial for selecting cultivars suited to unique environmental conditions. This study explores machine learning algorithms that leverage multispectral sensor data from UAVs to evaluate rice varieties. It focuses on three paddy rice types at different ages (six, nine, and twelve weeks after planting), analyzing data from four spectral bands and vegetation indices using various algorithms for classification. The results show that the neural network (NN) algorithm is superior, achieving an area under the curve value of 0.804. The twelfth week post-planting yielded the most accurate results, with green reflectance the dominant predictor, surpassing the traditional vegetation indices. This study demonstrates the rapid and effective classification of rice varieties using UAV-based multispectral sensors and NN algorithms to enhance agricultural practices and global food security.

**Keywords:** drone; neural network; precision agriculture; paddy; remote sensing

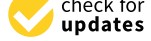



## 1. Introduction

Variety classification plays a pivotal role in assessing yield and optimizing rice-field cultivation. Different rice varieties possess distinctive characteristics, including growth habits [1–3], nutrient requirements [4–6], disease resistance [7–10], and tolerance to environmental stressors, such as drought, heat, and pests [11,12]. Identifying the best-performing varieties in each region allows farmers to enhance their yields and increase profitability. Accurate variety classification leads to more precise yield estimation. Additionally, diverse rice varieties require specific management practices, including fertilization and irrigation. Tailoring these practices to the particular needs of each variety, made possible by accurate variety prediction, enhances crop health and boosts yields. Variety classification is also crucial for predicting plant capabilities for handling threats such as pests and diseases [13]. Moreover, yield and pest/disease management are critical considerations in agricultural insurance assessments, underscoring the significance of variety classifications in the initial stages of such evaluations. For this purpose, researchers have made considerable efforts to develop remote sensing technology, including the utilization of drones or unmanned aerial vehicles (UAVs), which have made significant strides in agriculture [14–17].

With their remote sensing technology, UAVs have been growing rapidly in the last decade. They offer high spatial resolution for agriculture and flexible and low-cost monitoring, especially for frequent or scheduled monitoring [18–24]. In line with this, sensors for UAVs are also increasing [25,26]. The multispectral sensor is one of the sensors developed as a payload for UAVs, imitating the multispectral imager on satellite platforms, such as Thematic Mapper (TM) on Landsat 8 or Multispectral Instrument (MSI) on Sentinel 2, with more detail in spatial resolution. Multispectral sensors attached to UAV are utilized in agriculture primarily to calculate vegetation indices, such as the traditional normalized difference vegetation index (NDVI), to determine the conditions of plants. The NDVI derived from a multispectral sensor on a UAV is accurate, with high determination values compared to direct measurement [27]. The leaf chlorophyll index (LCI) is another vegetation index used to estimate leaf chlorophyll content, an indicator of photosynthetic activity. LCI relies on reflectance values from red-edge and near-infrared wavelengths, which are sensitive to the chlorophyll content in plant leaves [28–31]. Research indicates that the LCI has the potential to offer more accurate estimates of chlorophyll content than other indices such as the NDVI and the chlorophyll vegetation index (CVI) [27–31].

Existing studies have employed approaches such as extreme learning machine (ELM), support vector machine (SVM), and random forest (RF) algorithms [32,33] to accurately classify paddy growth stages and map the spatial distribution of paddy rice fields. These studies collectively suggest that the SVM and RF algorithms are suitable for classifying different paddy varieties using remote sensing data. However, the classification of paddy rice varieties using UAV data has not been extensively explored. Although machine learning techniques have been extensively studied for various topics, such as yield estimation, their application for rice variety classification using multispectral data from UAVs is still limited. Wang et al. [34] investigated hyperspectral imaging (HSI) to distinguish rice varieties and assess their quality, successfully generating a classification map that visualized distinct rice varieties. However, it is essential to note that the analysis was limited to the rice grains. Another study by Darvishsefat et al. [35] evaluated the spectral reflectance of rice varieties using hyperspectral remote sensing, demonstrating the potential for accurately mapping the cultivated areas of rice varieties based on hyperspectral remotely sensed data. Nevertheless, when considering the advantages of multispectral sensors, such as better performance, cost-effectiveness, and improved spatial coverage compared to hyperspectral sensors [36–38], the reduced complexity and data processing requirements of multispectral sensors outweigh the benefits of the higher spectral resolution offered by hyperspectral data. Hyperspectral remote sensing, while capturing a more detailed spectral signature, requires sophisticated data processing techniques and considerable computational resources. Consequently, there is an opportunity for further research on integrating machine learning algorithms with the multispectral data collected by UAVs for rice variety classification.

In this study, we employed a multispectral sensor attached to a quadcopter UAV to classify paddy varieties and explore machine learning applications for classifying them. We evaluated several machine learning algorithms, including a neural network (NN), decision tree, SVM, RF, naïve Bayes, and logistic regression. Additionally, we examined the effectiveness of ensemble algorithms, such as AdaBoost, gradient boosting, and a combination of high-performing algorithms as stacked learners. The performance of these algorithms was assessed by classifying three rice varieties, INPARI-32, INPARI-33, and INPARI-43 [39–41], which are high-yield rice varieties developed by the Indonesian Agency for Agricultural Research and Development (IAARD) in collaboration with the International Rice Research Institute (IRRI), at three different growth stages: six, nine, and twelve weeks after planting (WAP). This study aimed to determine the most effective algorithm for accurately classifying rice varieties, considering features such as reflectance from multispectral bands and vegetation indices. By enhancing automated systems for rice variety classification, this study contributes to improved agricultural practices. It offers

a reliable method for farmers, researchers, and stakeholders to efficiently identify and manage specific rice varieties at the optimal growth stage.

The structure of this article is as follows. The Introduction provides an overview of the research problem, its significance, and our research objectives. In the Methods section, we detail the methods and techniques employed in our study, including a description of the study location and instruments, data collection procedures, experimental setup, data analysis methods, feature selection, accuracy assessment procedures, and feature contribution analysis. We present the key findings of our research in the Results section and thoroughly discuss them in the Discussion section. Finally, we conclude our study and propose potential areas for improvement in the Conclusions and Future Work section.

## 2. Methods

This study involved a series of well-defined steps, beginning with the acquisition of field data and culminating in the preparation of these data for use as a dataset in machine learning algorithms. The overall progression of these steps is visually represented in Figure 1, which provides an overview of the workflow of the study. Each part of the general flow is explained in the following section.

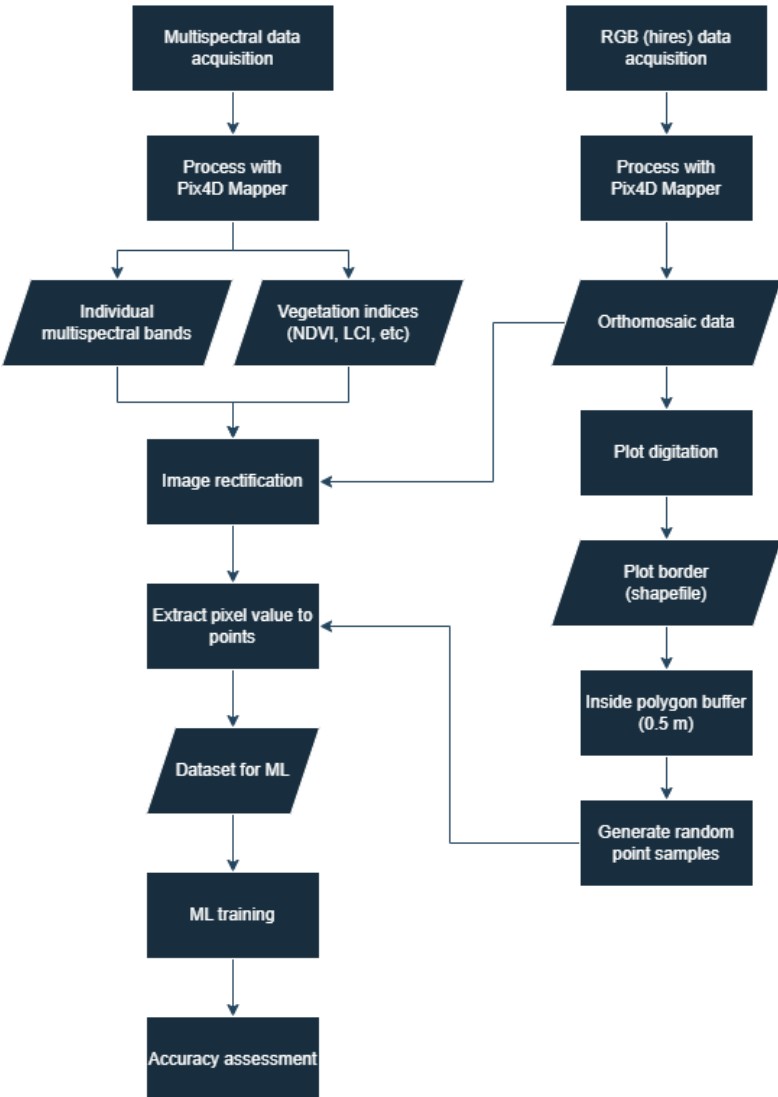

**Figure 1.** General flow of the study.

### 2.1. Location and Instruments

This study was conducted at the Indonesia Center for Rice Research in Subang, West Java Province, Indonesia (Figure 2). Two quadcopter drones were used: DJI Mavic 2 Pro and DJI Inspire 1 produced by DJI (SZ DJI Technology Co., Ltd., Shenzhen, Guangdong, China). The former was used to generate a detailed map of the study area. A multispectral sensor called Parrot Sequoia was attached as a payload on the latter (DJI Inspire 1) (Figure 3a).

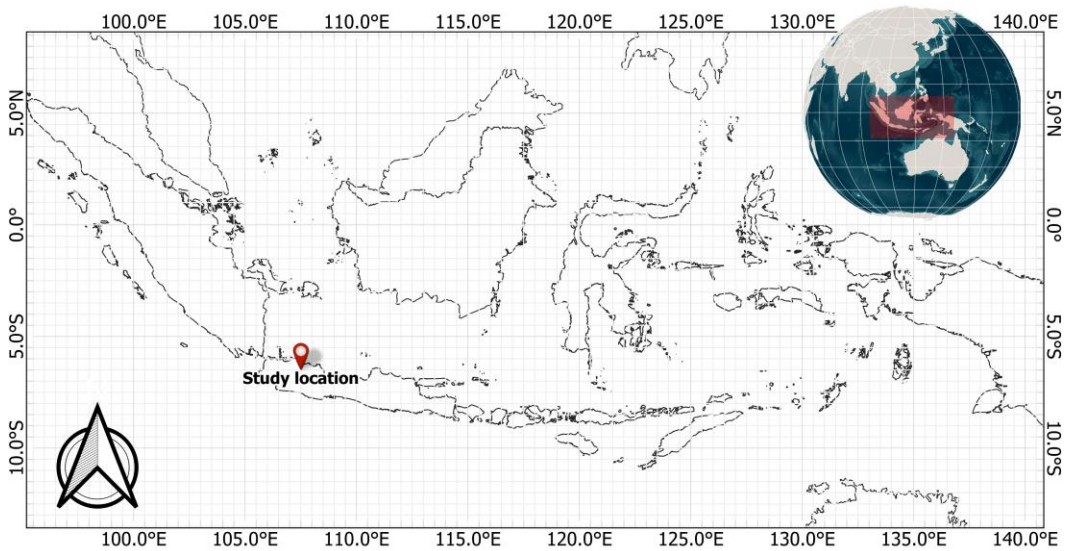

**Figure 2.** Map depicting study location at Subang (red dot), West Java Province, Indonesia (red rectangle in the overview map).

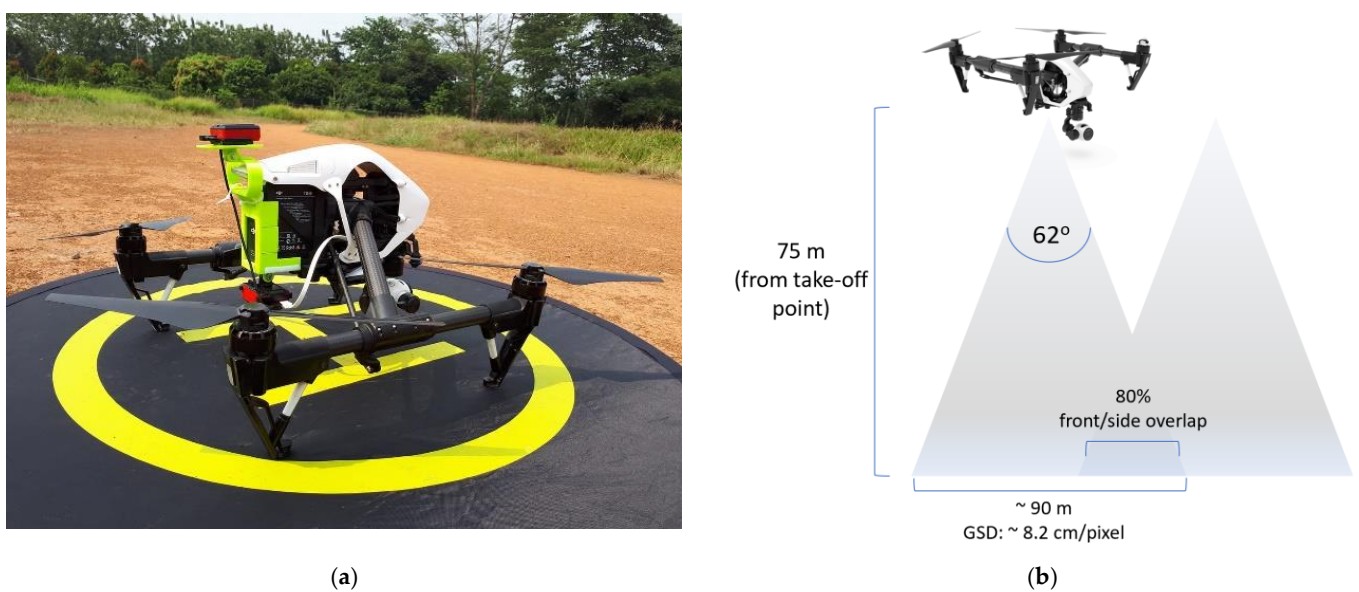

(**a**)           (**b**)

**Figure 3.** DJI Inspire 1 with Parrot Sequoia attached (**a**) and illustration of flight planning (**b**).

Parrot Sequoia is equipped with a 4-band multispectral sensor with 1.2 megapixels (1280 × 960). The four bands were green (550 ± 40 nm), red (660 ± 40 nm), red edge (735 ± 10 nm), and near-infrared (NIR) (790 ± 40 nm) with horizontal and vertical fields of view (FOV) of 62° and 49°, respectively. An attached sunshine sensor compensates for the dynamic changes in solar illumination. This sensor makes it possible to obtain the reflectance under all illumination conditions [42]. The sensor was calibrated using a calibration panel to normalize the images and correct for atmospheric effects.

The UAV was operated at an altitude of 75 m from the takeoff point to capture the study area, as shown in Figure 3b. This flight altitude satisfied safety precautions, considering the presence of structures and trees near the study plots while ensuring a detailed ground sampling distance. We used Pix4d Capture (Pix4D S.A, Prilly, Switzerland) version 4.13.0 as the mission planning software and configured the side and front overlap fractions to 80%. Using the "fast mode" in the Pix4D Capture, the UAV captured the images while translating with a constant velocity of approximately 6 m/s. With a horizontal FOV of 62°, the width of the captured area was approximately 90 m and the ground sampling distance was 8.2 cm/pixel (Figure 3b). Because this resolution was satisfactory enough for the purpose of this study, we did not further assess the effects of different flight altitudes. The observations took place from 09:00 to 11:00 on 9 November, 30 November, and 28 December 2022, under clear sky weather conditions.

## 2.2. Experimental Plot Design

In this study, 24 experimental plots were observed, each featuring one of three high-yield rice varieties: INPARI-32, INPARI-33, and INPARI-43. These three varieties share similar physical characteristics, with the primary distinguishing feature being height. INPARI-32 was the tallest, measuring approximately 97 cm, followed by INPARI-33 at approximately 93 cm and INPARI-43 at approximately 83 cm.

Each individual variety is represented by eight repetition plots. These plots were distributed across two distinct locations: Unit 1 and Unit 2 (Figure 4). Plots in Unit 1 measured 12 m by 4 m, while plots in Unit 2 measured 9.5 m by 4.2 m. In the present analysis, no distinction was made between these two locations. Twenty random points were created in each plot. To ensure that the points were precisely on the observed plants and not on the edge, a 0.5 m buffer was kept at each border to generate random points. This equated to 20 random points multiplied by 24 plots, resulting in 480 samples (Table 1). From these samples, we extracted all multispectral bands and indices associated with each point. We observed plants at three different growth stages: six, nine, and twelve WAP.

**Table 1.** Sampling points.

| Unit | Varieties | Number of Repetitions | Sample Points | Total |
|------|-----------|-----------------------|---------------|-------|
| 1 | INPARI-32 | 4 | 20 | 80 |
| | INPARI-33 | 4 | 20 | 80 |
| | INPARI-43 | 4 | 20 | 80 |
| 2 | INPARI-32 | 4 | 20 | 80 |
| | INPARI-33 | 4 | 20 | 80 |
| | INPARI-43 | 4 | 20 | 80 |
| | Total | | | 480 |

## 2.3. Airborne Data Processing

In addition to the multispectral bands, we generated vegetation indices (VIs) from the acquired data. These indices, derived from mathematical formulas combining different spectral bands, offer further insight into the health, vigor, and physiological characteristics of rice plants. Table 2 summarizes the formulas used to generate the vegetation indices. The raw data acquired from the airborne platform in the field were processed using the Pix4D Mapper software version 4.8.4 to obtain orthomosaic data for each multispectral band. Subsequently, six vegetation indices, NDVI, LCI, GNDVI (green NDVI), SAVI (soil adjusted vegetation index), OSAVI (optimized SAVI), and LAI (leaf area index), were derived from the orthomosaic multispectral data. To ensure precise alignment of the data, we performed a geometric correction using a map derived from high-resolution data obtained from DJI Mavic 2. In this case, the orthomosaic map from DJI Mavic 2 was used as a reference. Both the individual multispectral bands and vegetation indices were subsequently orthorectified using an orthomosaic map. We used several points within the field, such as the corner of the block, as ground control points (GCP).

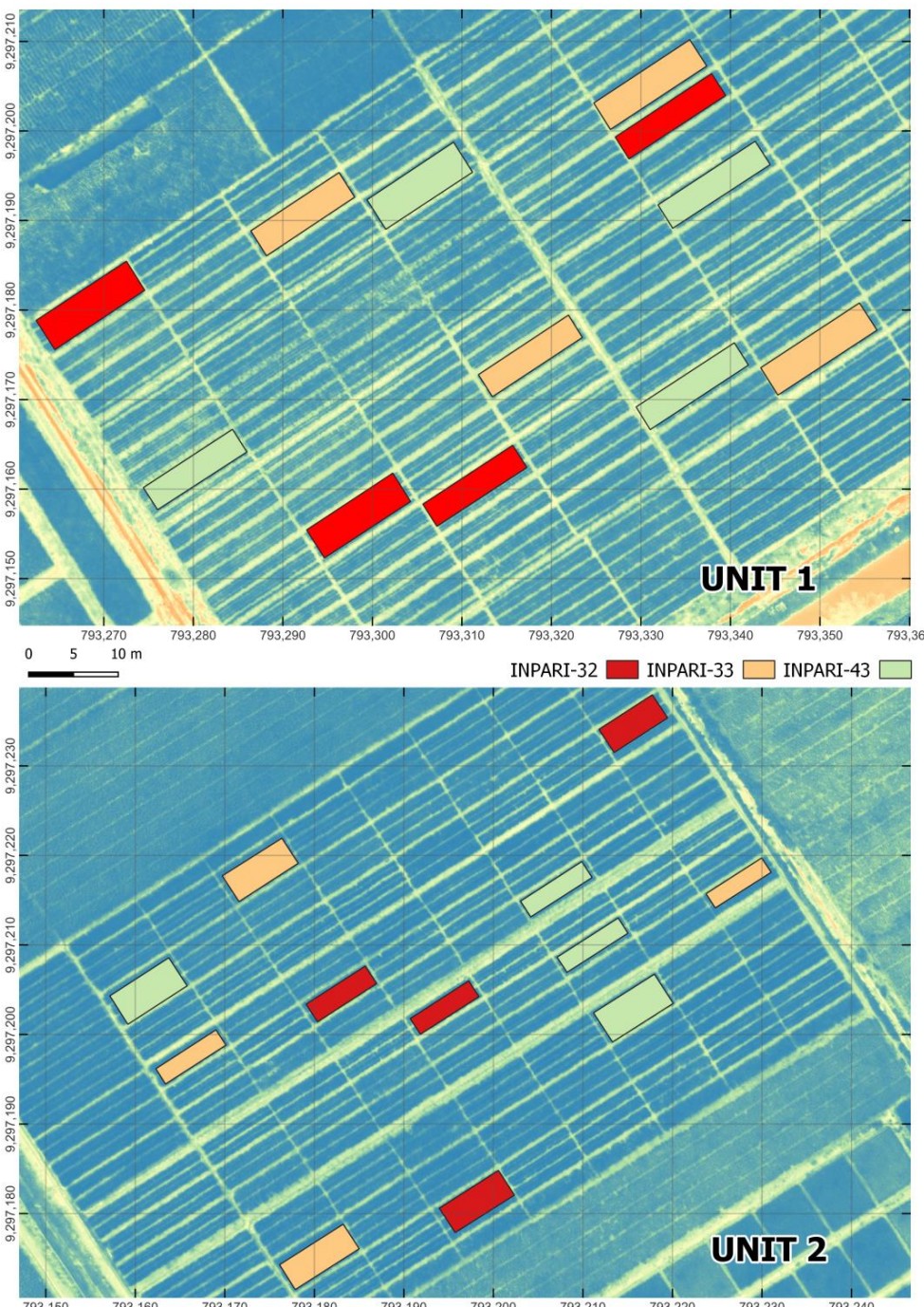

**Figure 4.** Experimental plots used in this study. There were 24 plots at two locations (Unit 1 and Unit 2), with 12 plots in each plot and four plots for each variety in a single location.

**Table 2.** Formulas for generating vegetation indices used as features.

| VI Name | Formula | Reference |
|---------|---------|-----------|
| NDVI | NDVI = (NIR − Red)/(NIR + Red) | [43,44] |
| LCI | LCI = (NIR − RedEdge)/(NIR + RedEdge) | [43] |
| GNDVI | GNDVI = (NIR − Green)/(NIR + Green) | [45] |
| SAVI | SAVI = ((NIR − Red)/(NIR + Red + 0.5)) × 1.5 | [46] |
| OSAVI | OSAVI = ((NIR − Red)/(NIR + Red + 0.16)) × (1 + 0.16) | [47] |
| LAI | LAI = [ln(NIR/Red)/(1.4 × (NIR/Red))] − 1 | [48] |

### 2.4. Feature Selection

We utilized a range of scoring methods to assess the relationship between the features (multispectral bands and vegetation indices) and each target variable (INPARI-32, INPARI-33, and INPARI-43). These methods encompass internal scorers, such as information gain, chi squared, and linear regression. By measuring the correlation between the variables and the target variable, we assigned scores to each feature by employing the appropriate scorers and models to determine their significance and predictive capabilities. At each growth stage, we conducted rankings and selected the top five features for that specific stage.

### 2.5. Classification Algorithms

We conducted an extensive evaluation of seven different machine learning classification algorithms to assess their performance in classifying rice varieties. The algorithms we tested included NN, decision tree, SVM, RF, naïve Bayes, and logistic regression. In addition to these individual algorithms, we also explored the performance of ensemble algorithms, which combine multiple base classifiers to improve classification performance. Our study specifically examined AdaBoost, gradient boosting, and a stack of high-performance algorithms. Ensemble algorithms have been shown to enhance the classification accuracy by combining the strengths of different models and mitigating their weaknesses. Table 3 lists all classification algorithms and their respective references.

**Table 3.** Classification algorithms used in this study.

| Algorithm | Reference |
|---|---|
| AdaBoost | [49] |
| Gradient boosting | [50] |
| Logistic regression | [51] |
| Naïve Bayes | [52] |
| NN | [53] |
| RF | [54] |
| Stack algorithm | [55] |
| SVM | [56] |
| Decision tree | [57] |

### 2.6. Accuracy Assessment

K-fold cross-validation is a method for evaluating machine learning models. In the present analysis, we divided the dataset into ten subsets, trained the model on nine of them, and tested it on the remaining one. This process was repeated ten times, with each subset serving as the validation set. It provides a robust estimate of the model's performance, helping to detect overfitting and make efficient use of available data. We conducted accuracy assessments using five standard parameters: area under the curve (AUC), classification accuracy (CA), F1, precision, and recall. These parameters can provide a comprehensive measure of the effectiveness of algorithms in accurately classifying rice varieties.

All classifications, as well as the accuracy assessment, were conducted using Orange Data Mining [58] software version 3.34. This open-source software is used for data mining and machine learning. Developed by the Bioinformatics Lab at the University of Ljubljana, Slovenia and a community of developers and researchers worldwide, it allows users to build workflows and models using drag-and-drop components. In this study, we used the default parameters set in the software, without parameter tuning.

### 2.7. Feature Contribution Analysis

During the analysis, the performance of the classification model was tested with and without individual features (each of the input variables, namely, four multispectral bands and six vegetation indices) to observe how each feature affects the accuracy of the classification. The contribution of each feature is then measured using the permutation

feature importance technique. This is a method for understanding which features are most influential in making predictions, and helps in feature selection and model interpretation. The basic idea behind permutation feature importance is to measure the extent to which a model's performance (e.g., accuracy, F1 score, or any other relevant metric) decreases when the values of a particular feature are randomly shuffled while keeping other features constant. To investigate how each feature contributes to the classification of a specific class, we used the SHAP library [59], which provides an estimation of the degree to which each feature influences the output of the model.

## 3. Results

### 3.1. Spectral Characteristics of Observed Plants

We observed dynamic changes in the spectra and vegetation indices values from six to 12 WAP. Figure 5 shows the dynamic changes in spectral reflectance and vegetation indices for all three varieties. The spectral values of plants tend to change during the growing stages [45,60] because of the physiological and morphological changes that occur as the plant develops. Under normal conditions, the paddy spectra change from six to nine WAP. This is because nine WAP are at the peak of the vegetative stage of paddy growth [61]. The values of NIR and other vegetation indices (NDVI, LCI, GNDVI, SAVI, and OSAVI) usually decrease from nine WAP to twelve WAP because the latter is the period of the generative stage when the plants are ready to be harvested [62]. Similar results were observed in the present study. The spectral reflectance of the red-edge and NIR, as well as the vegetation indices, increased from six to nine WAP and decreased from nine to twelve WAP.

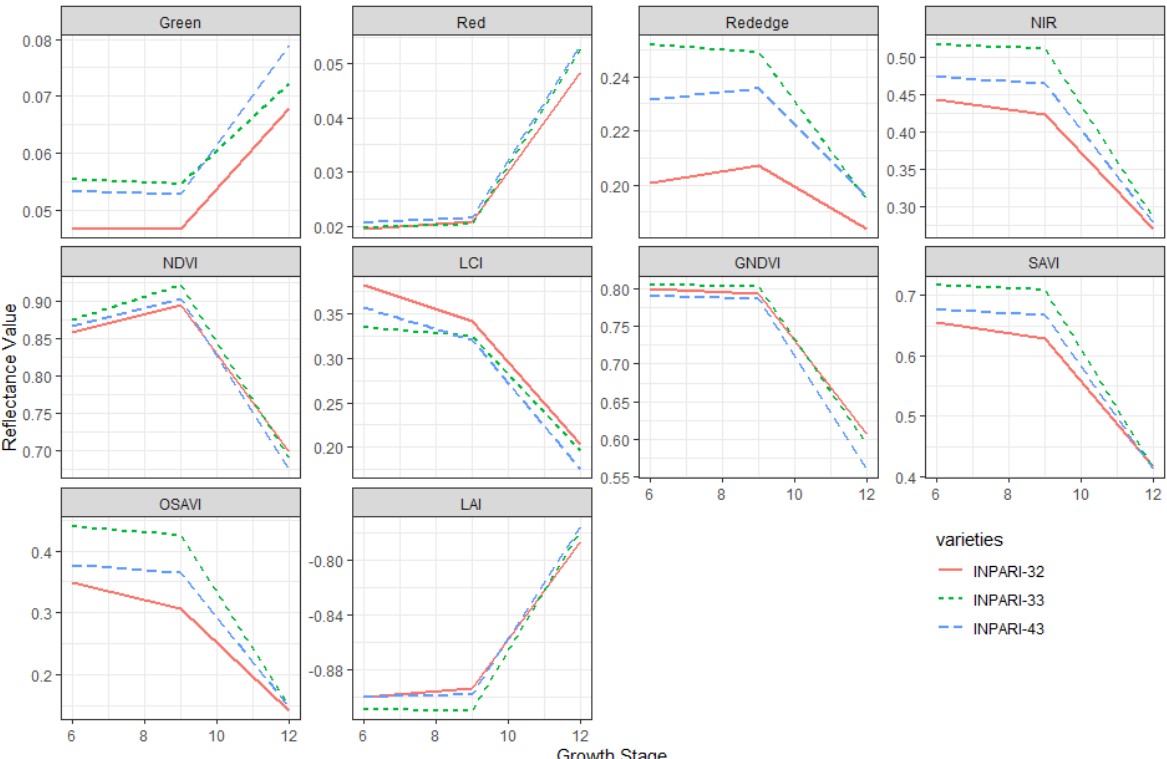

**Figure 5.** Dynamic change in average spectral reflectance and VIs during growth-stage of observed plants from six WAP until twelve WAP.

### 3.2. Feature Selection

Table 4 offers a succinct yet comprehensive summary of our feature selection analysis, providing a detailed representation of the specific features selected along with their associated attributes. Notably, our results indicate that the Green feature was the sole variable that consistently appeared across all growth-stage datasets.

**Table 4.** Summary of feature selection (O = remain, X = dropped).

| Feature | Six WAP | Nine WAP | Twelve WAP |
| --- | --- | --- | --- |
| NIR | O | O | X |
| Green | O | O | O |
| Red | O | X | X |
| Red edge | O | O | X |
| NDVI | X | X | O |
| LCI | X | X | O |
| GNDVI | X | X | O |
| SAVI | X | O | X |
| OSAVI | O | O | X |
| LAI | X | X | O |

After identifying the features for each growth stage, we assessed the intraclass variability of these features. Intraclass variability refers to the degree of variation within each class or category of rice varieties and is evaluated as the standard deviation. This analysis provides insights into the consistency and reliability of the features for differentiating between rice varieties. A lower standard deviation suggests that the values of the feature within a specific growth stage are more closely clustered around the mean, whereas a higher standard deviation indicates more noticeable variability within the class. By quantifying the intraclass variability, we assessed the discriminative power of the features and their ability to accurately distinguish between different rice varieties. Lower intraclass variability implies that the feature is more consistent and reliable in capturing the distinctive characteristics of each variety, thereby enhancing the precision of the classification process. From this analysis, we found that the Green feature exhibited the lowest standard deviation among the growth stages (Figure 6).

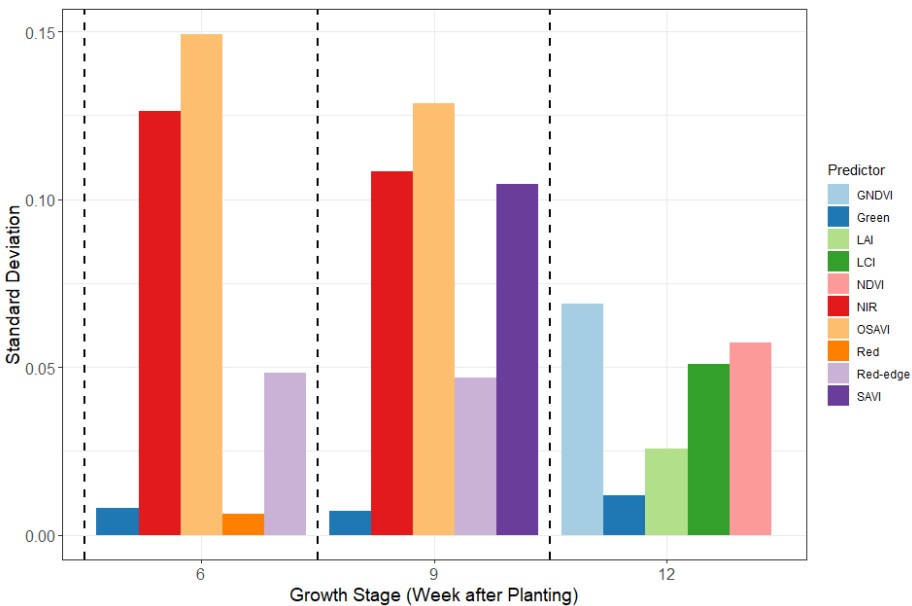

**Figure 6.** Standard deviation (SD) of the top five selected features for each growth stage, highlighting twelve WAP as the period with the narrowest distribution.

### 3.3. Varieties Classification

We observed the possibility of using multiple features to determine the rice varieties. As we assessed nine classification algorithms with four multispectral bands and six vegetation indices as features, the performance of each feature was observed in accordance with the classification algorithm.

Table 5 presents the average performance of various machine learning algorithms for classifying paddy rice varieties using multiple variables as features. Upon analyzing the results, it became apparent that the performance of the algorithms varied based on the growth stage of the plants. For example, the best algorithm at six WAP was the NN; at nine WAP, it was the stack algorithm; and at twelve WAP, it was the NN. Among these, the NN algorithm demonstrated the highest performance in terms of CA, F1, precision, and recall, specifically at twelve WAP.

**Table 5.** Average performance of seven machine learning algorithms over classes. A single asterisk (*) indicates the highest value within the same growth stage and a double asterisk (**) indicates the highest value across all growth stages.

| Growth Stage (WAP) | Algorithm | AUC [1] | CA [2] | F1 [3] | Precision | Recall |
|---|---|---|---|---|---|---|
| 6 | AdaBoost | 0.648 | 0.530 | 0.535 | 0.545 | 0.530 |
| | Gradient boosting | 0.792 * | 0.602 | 0.598 | 0.596 | 0.602 |
| | Logistic regression | 0.634 | 0.458 | 0.518 | 0.417 | 0.458 |
| | Naïve Bayes | 0.722 | 0.417 | 0.500 | 0.499 | 0.517 |
| | NN | 0.790 | 0.618 * | 0.612 * | 0.608 * | 0.618 * |
| | RF | 0.762 | 0.571 | 0.569 | 0.568 | 0.571 |
| | Stack algorithm | 0.786 | 0.608 | 0.597 | 0.592 | 0.608 |
| | SVM | 0.554 | 0.357 | 0.344 | 0.346 | 0.357 |
| | Decision tree | 0.662 | 0.533 | 0.531 | 0.533 | 0.533 |
| 9 | AdaBoost | 0.590 | 0.454 | 0.455 | 0.458 | 0.454 |
| | Gradient boosting | 0.696 | 0.508 | 0.503 | 0.500 | 0.508 |
| | Logistic regression | 0.595 | 0.457 | 0.405 | 0.417 | 0.457 |
| | Naïve Bayes | 0.700 | 0.517 | 0.509 | 0.508 | 0.517 |
| | NN | 0.750 * | 0.590 | 0.580 | 0.579 | 0.590 |
| | RF | 0.704 | 0.549 | 0.545 | 0.542 | 0.549 |
| | Stack algorithm | 0.743 | 0.600 * | 0.581 * | 0.586 * | 0.600 * |
| | SVM | 0.649 | 0.429 | 0.412 | 0.423 | 0.429 |
| | Decision tree | 0.682 | 0.571 | 0.567 | 0.567 | 0.571 |
| 12 | AdaBoost | 0.662 | 0.550 | 0.550 | 0.550 | 0.550 |
| | Gradient boosting | 0.802 | 0.625 | 0.622 | 0.621 | 0.625 |
| | Logistic regression | 0.548 | 0.372 | 0.363 | 0.358 | 0.372 |
| | Naïve Bayes | 0.717 | 0.506 | 0.494 | 0.490 | 0.506 |
| | NN | 0.804 ** | 0.644 ** | 0.642 ** | 0.642 ** | 0.644 ** |
| | RF | 0.788 | 0.628 | 0.626 | 0.626 | 0.628 |
| | Stack algorithm | 0.800 | 0.641 | 0.636 | 0.635 | 0.637 |
| | SVM | 0.646 | 0.359 | 0.368 | 0.387 | 0.359 |
| | Decision tree | 0.666 | 0.528 | 0.532 | 0.537 | 0.528 |

[1] AUC: area under the curve; [2] CA: classification accuracy; [3] F1: F1 score.

We further assessed the classification performance of the seven machine learning algorithms for each rice variety, as shown in Table 6. We found that for six and nine WAP, the stack algorithm exhibited commendable performance in terms of CA when applied to the INPARI-32 variety. Notably, this algorithm achieved a CA of 0.834 and 0.816 for six and nine WAP, respectively, effectively distinguishing INPARI-32 from the other varieties. However, the results differed slightly when considering the results for twelve WAP. During this period, the algorithms demonstrated a more satisfactory ability to differentiate INPARI-43 from other varieties. Specifically, NN yielded the highest CA of 0.841, closely followed by the stack algorithm, with a CA of 0.831.

The performance of each algorithm is reflected in the classification of the paddy varieties. As shown in Figure 7, NN performed well in classifying the three varieties: INPARI-32, INPARI-33, and INPARI-43. Approximately 60% of the training samples for INPARI-32 were correctly classified. The remaining 40% were classified incorrectly as INPARI-33 and INPARI-43. Logistic regression provided the lowest accuracy. This algorithm failed to classify the rice varieties correctly.

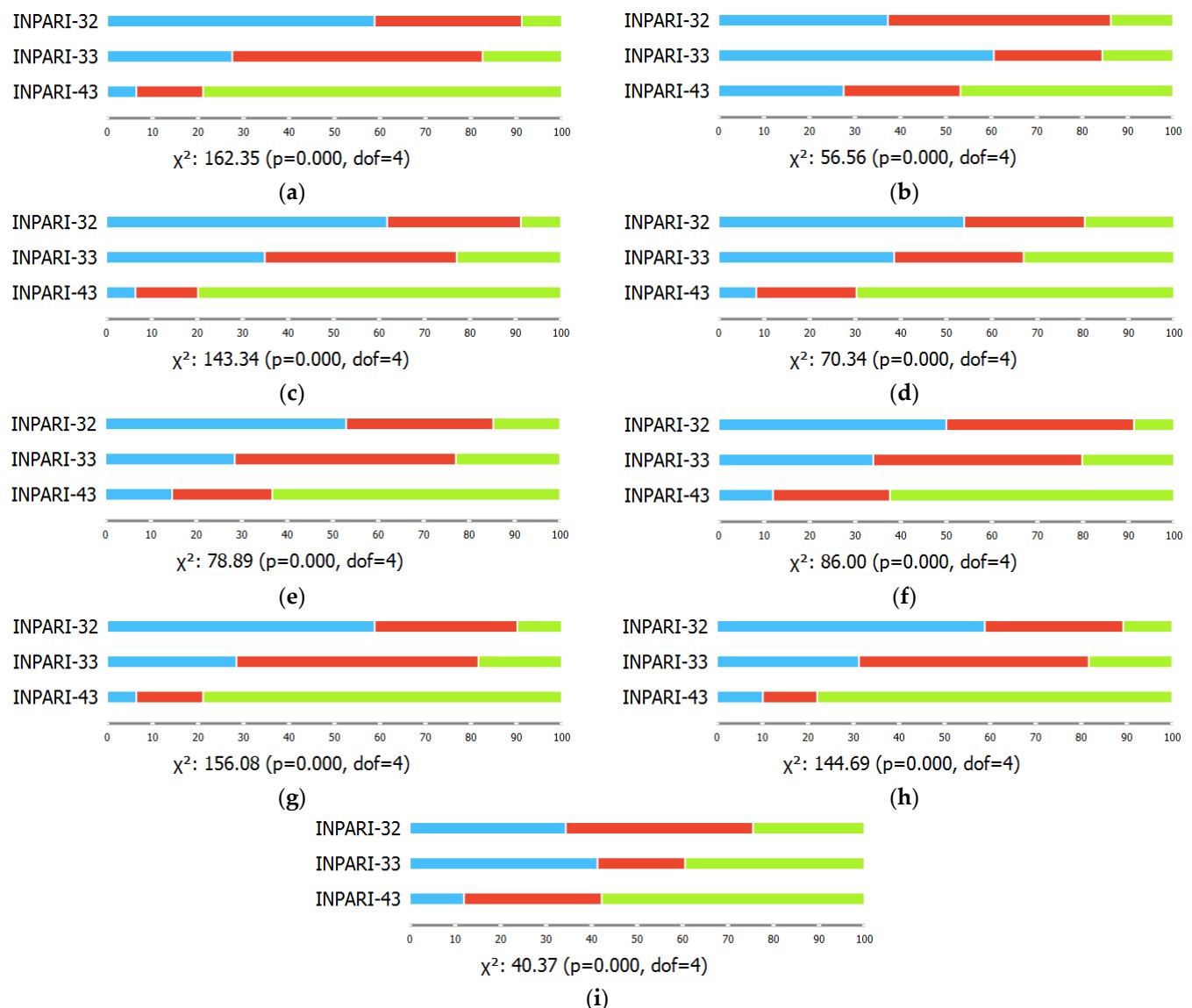

**Figure 7.** Box plots depicting the performance of seven algorithms to determine three paddy rice varieties (blue bar: INPARI-32; red bar: INPARI-33; green bar: INPARI-43) using the data for twelve WAP: (**a**) NN, (**b**) SVM, (**c**) RF, (**d**) Naïve Bayes, (**e**) AdaBoost, (**f**) decision tree, (**g**) stack algorithm, (**h**) gradient boosting, and (**i**) logistic regression.

*3.4. Features Contribution*

Based on these results, the NN algorithm applied during the twelfth WAP emerged as the most effective algorithm for determining rice varieties in this study. Subsequently, we conducted an analysis to assess the contribution of each feature to identifying the varieties. By permutation feature importance, we discovered that among the selected features from the twelfth week after planting, the Green feature exhibited the highest level of contribution, surpassing vegetation indices such as the LCI, GNDVI, NDVI, and LAI. This significance of the Green feature is evident in Figure 8, where it is considered the most essential feature. Notably, the absence of this band resulted in a decrease of approximately 0.24 in the AUC of the classification. A decrease in the AUC typically indicates a reduction in the model's ability to distinguish between classes. Since the AUC values range from 0 to 1, a decrease of 0.24 in the AUC is significant, suggesting that the Green feature has a substantial impact on the model's performance.

**Table 6.** Performance of the seven machine learning algorithms for classifying each rice variety in terms of classification accuracy. A single asterisk (*) indicates the highest value within the same variety and the same growth stage, while a double asterisk (**) indicates the highest value across all varieties in the same growth stage.

| Growth-Stage (WAP) | Algorithm | INPARI-32 | INPARI-33 | INPARI-43 |
|---|---|---|---|---|
| 6 | AdaBoost | 0.781 | 0.671 | 0.608 |
| | Gradient boosting | 0.824 | 0.740 | 0.639 |
| | Logistic regression | 0.655 | 0.652 | 0.608 |
| | Naïve Bayes | 0.743 | 0.665 | 0.627 |
| | NN | 0.828 | 0.759 | **0.649 *** |
| | RF | 0.815 | 0.708 | 0.618 |
| | Stack algorithm | **0.834 ** ** | **0.743 *** | 0.639 |
| | SVM | 0.542 | 0.624 | 0.549 |
| | Decision tree | 0.777 | 0.680 | 0.608 |
| 9 | AdaBoost | 0.705 | 0.625 | 0.578 |
| | Gradient boosting | 0.765 | 0.654 | 0.597 |
| | Logistic regression | 0.679 | 0.603 | 0.632 |
| | Naïve Bayes | 0.721 | 0.689 | 0.625 |
| | NN | 0.797 | **0.721 *** | 0.663 |
| | RF | 0.781 | 0.683 | 0.635 |
| | Stack algorithm | **0.816 ** ** | 0.714 | 0.670 |
| | SVM | 0.651 | 0.641 | 0.565 |
| | Decision tree | 0.778 | 0.686 | **0.679 *** |
| 12 | AdaBoost | 0.703 | 0.647 | 0.750 |
| | Gradient boosting | 0.728 | **0.694 *** | 0.828 |
| | Logistic regression | 0.609 | 0.491 | 0.644 |
| | Naïve Bayes | 0.694 | 0.597 | 0.722 |
| | NN | **0.753 *** | **0.694 *** | **0.841 ** ** |
| | RF | 0.750 | 0.684 | 0.828 |
| | Stack algorithm | 0.750 | 0.688 | 0.831 |
| | SVM | 0.500 | 0.497 | 0.722 |
| | Decision tree | 0.684 | 0.597 | 0.775 |

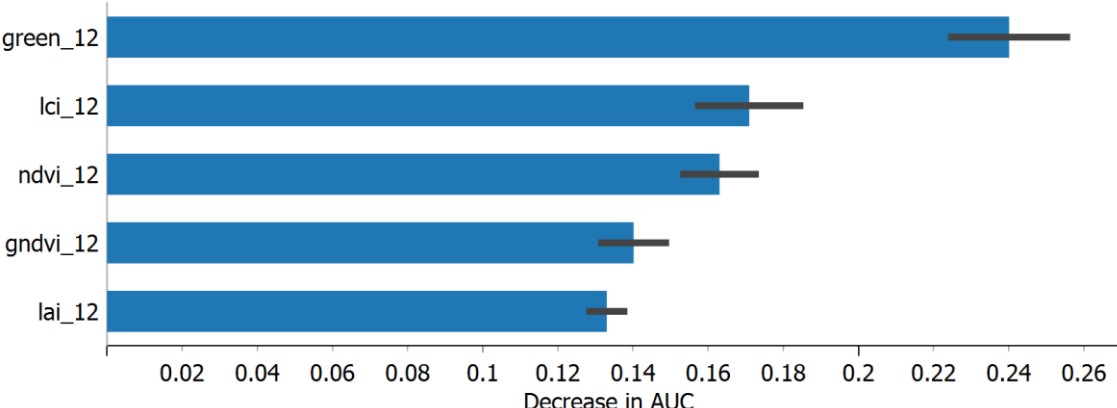

**Figure 8.** The permutation feature importance of five features during the twelve WAP showed Green reflectance as an essential feature.

We also analyzed the contribution of each feature to the classification of each rice variety. The results indicated that the Green feature demonstrated the highest predictive power in classifying the INPARI-32 variety, as illustrated in Figure 9a. Specifically, a lower value (reflectance) of Green during the twelfth WAP exhibited a greater contribution towards identifying INPARI-32. By contrast, a higher LCI value was observed to have a stronger association with the classification of this variety.

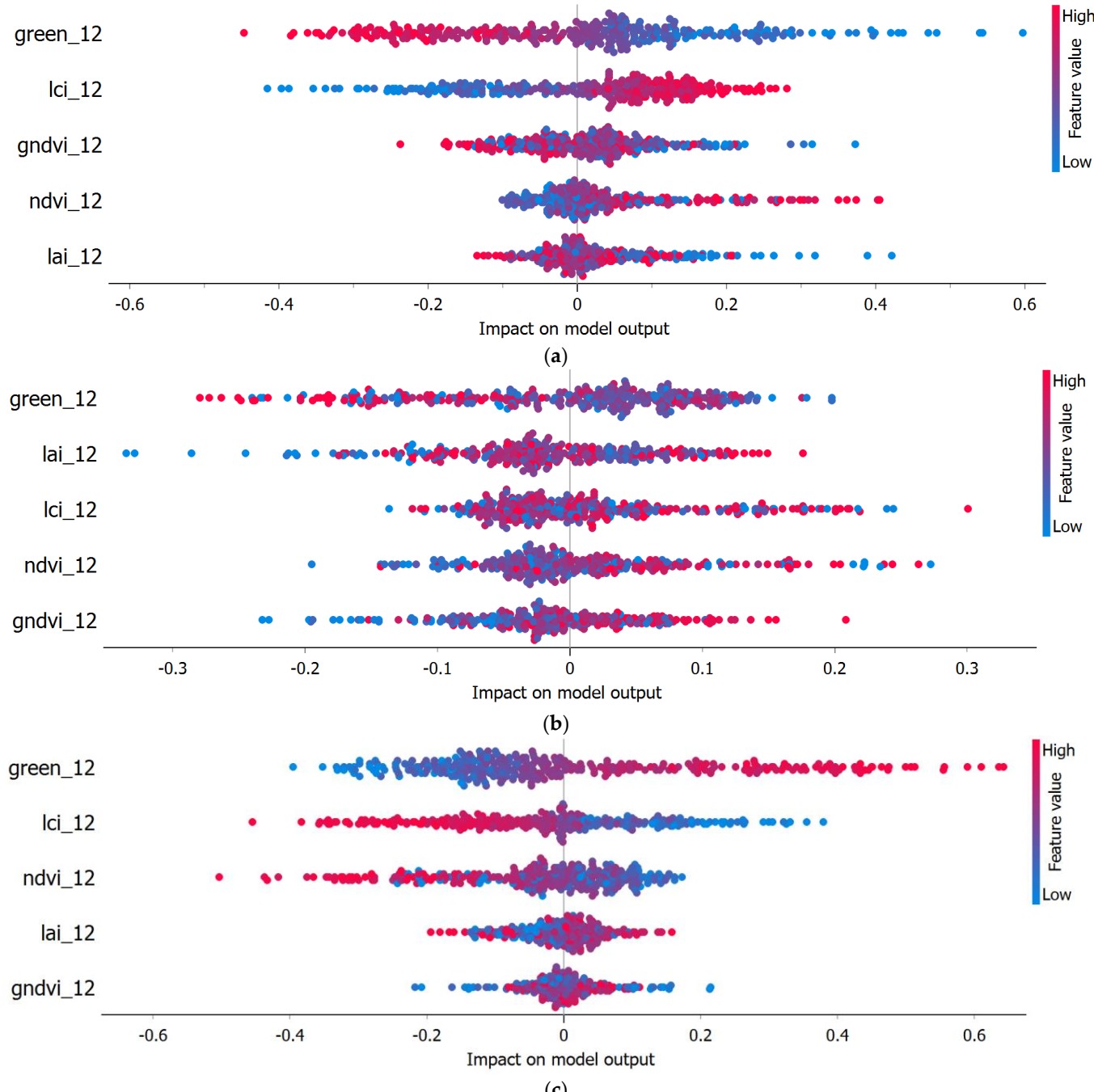

**Figure 9.** Features contributing to the detection of INPARI-32 (**a**), INPARI-33 (**b**), and INPARI-43 (**c**) using a neural network.

To detect INPARI-33, Green was also found to have a higher contribution, with higher values contributing more to the classification of this rice variety, as illustrated in Figure 9b. Additionally, Green was the feature with the highest contribution in classifying INPARI-43, followed by the two vegetation indices (LCI and NDVI), as shown in Figure 9c.

In this study, the data for the three varieties during the twelve WAP exhibited different reflectance characteristics from the Green reflectance, as shown by the normal distribution chart in Figure 10. INPARI-32 can be easily determined using the low reflectance of the green light. On the other hand, the high reflectance of green can be used to determine INPARI-43. Moreover, the map in Figure 11 shows a visual representa-

tion of Green reflectance from the experimental plots of the three varieties, which can be distinguished visually.

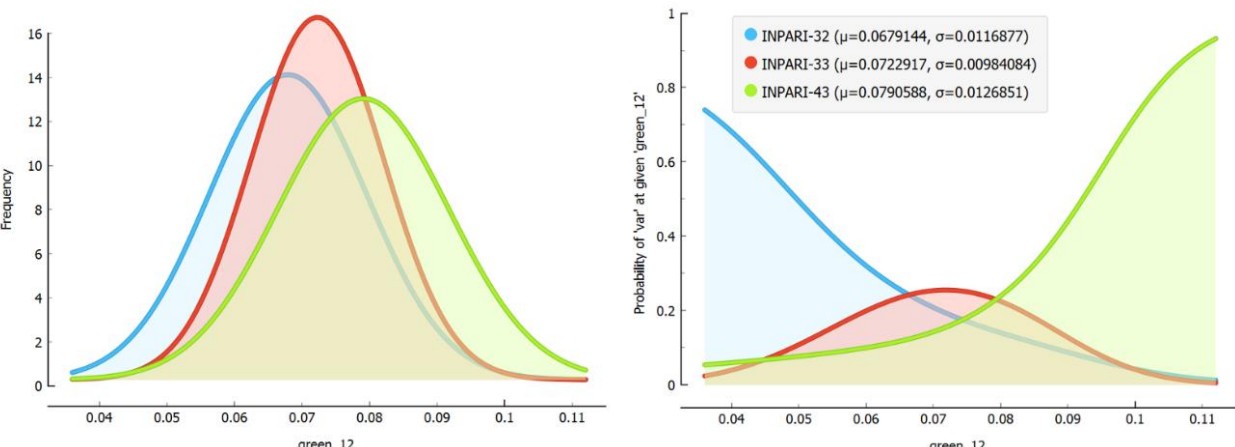

**Figure 10.** A normal distribution and its probability chart display varieties that are distinguishable through Green reflectance for the data in twelve WAP.

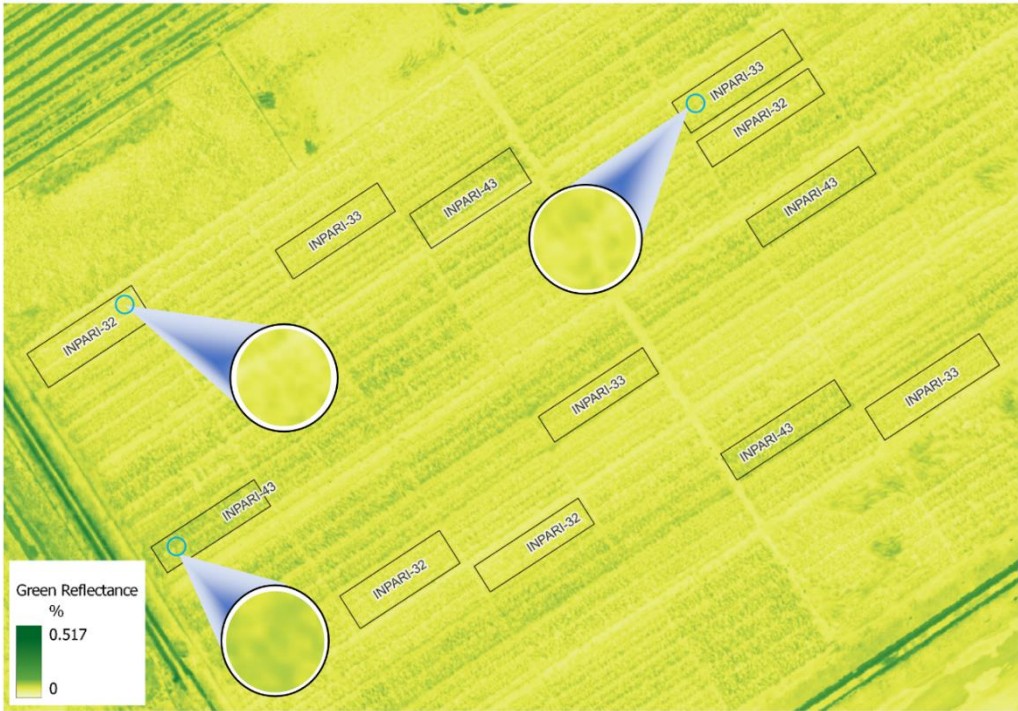

**Figure 11.** During the period of twelve WAP, three varieties could be visually distinguished using Green reflectance. In particular, INPARI-43, which has high Green reflectance, dominated the field.

## 4. Discussion

Every remote sensing technology has its advantages and disadvantages, and there are no exceptions when considering UAVs as the primary tool for monitoring rice varieties, as reported in this study. The most notable limitation is the limited capacity of UAVs to cover extensive areas. Their restricted spatial coverage may require additional flight missions, thereby creating logistical complexities in dealing with large areas. Satellite remote sensing provides a solution to address these challenges because it offers broad spatial coverage. Moreover, some satellite imagery already has a very high spatial resolution, such as GeoEye-1, developed and managed by the European Space Agency (ESA), and included in the WorldView constellation, which consists of four WorldView satellites and GeoEye-1.

It has a spatial resolution of 41 cm/pixel, which is almost equivalent to that of a UAV. In terms of the temporal resolution, which is related to the revisit time, it offers a 4.6-day temporal resolution, which is quite high. However, one major drawback of high-resolution satellite imagery is its unaffordability owing to its high cost. Hence, UAVs remain superior in terms of spatial, spectral, and temporal resolution, even though they have limited spatial coverage. Furthermore, the development of UAVs has now entered the era of fixed-wing UAVs, which offers more efficient and broader coverage.

This study investigated the application of machine learning to classify different paddy rice varieties using UAV data. Our analysis involved exploring various algorithms to identify the most effective approach tailored to this task. We also aimed to identify the most suitable growth stage for rice plants for a reliable variety classification. Our findings revealed that machine learning algorithms perform differently depending on the growth stage of the plant, with twelve WAP found to be the most optimum growth stage. Overall, the NN emerged as the most effective algorithm in this study.

Several studies have indicated that the growth stage of paddy rice plants affects transpiration, photosynthesis, and the response to soil water potential, as well as the impact of nitrogen application on yield [63,64]. In this study, we examined rice images captured at six, nine, and twelve WAP. During these stages, paddy rice undergoes significant physiological and morphological changes, which can affect its spectral reflectance. Our study revealed that twelve WAP is the optimum growth stage for rice variety classification, with NN as the best machine learning algorithm. This is due to several factors, such as morphological differences, increased canopy development, stable feature representation, and reduced intraclass variability. By the twelfth WAP, paddy rice plants have reached a more advanced growth stage, and there are more pronounced morphological differences among different plant varieties. These differences can include variations in leaf shape, size, color, and overall plant structure. These characteristics reflect the differences in the spectral reflectance captured by the UAV sensor. Machine learning algorithms can leverage these distinctive features to classify different varieties more accurately. It is also worth noting that the intraclass variability of the twelfth week after planting was the lowest, as shown in Figure 6. Such a reduction in intraclass variability plays a crucial role in improving classification accuracy, as it minimizes the overlap and confusion between different plant varieties. By reducing the variations within each class and enhancing the separability between classes, machine learning algorithms can achieve higher accuracy and reliability in the classification process. This observation is supported by previous studies, which highlighted the positive impact of reduced intraclass variability on classification performance. Fusheng et al. [65] reduced intraclass variability through instance-level embedding adaptation, significantly improving the classification accuracy of few-shot learning tasks. Hence, the combination of leveraging distinctive features and reduced intraclass variability at the twelve WAP contributes to the improved accuracy of machine learning algorithms in classifying plant varieties.

In line with the findings of this study, previous research by Tan et al. [66] also supported the superiority of the NN algorithm. They demonstrated that NNs outperformed maximum likelihood in land cover classification using Landsat multispectral data. Similarly, Etheridge et al. [67] conducted a classification study based on overall error rate. They found that the probabilistic NN exhibited the highest level of reliability, followed by the backpropagation and categorical learning networks. These findings highlight the consistent success of NN-based approaches for different classification tasks.

Other studies have explored the effectiveness of NNs in various agricultural applications. For example, Senan et al. [68], Bouguettaya et al. [69–71], and Ramesh and Vydeki [72] utilized deep learning techniques based on a convolutional neural network (CNN) for paddy leaf disease classification. Muthukumaran et al. [73], Amaratunga et al. [74], and Abdullah et al. [75] performed paddy yield prediction and forecasting using artificial NNs (ANNs). Although these studies did not specifically investigate the classification of rice

varieties, the successful utilization of NNs in related agricultural tasks further emphasizes the efficiency of these algorithms.

NNs are suitable for classification because of their efficiency, reliability, and accuracy, surpassing those of mainstream methods. Despite longer training times, the precise outcomes outweigh this drawback. Our study noted that the NN's extended training times were still faster than those of some ensemble algorithms. The NNs required approximately 3 s, while the others required less than 0.2 s. Therefore, it is important to consider increased training time when applying this algorithm. Their unique ability to handle multi-classification tasks and adapt through backpropagation sets them apart, excelling in intricate classification of paddy rice varieties. Despite the potential drawbacks of training time, NN accuracy and adaptability make it the ultimate choice for paddy rice classification.

In this study, we explored the predictive power of various features for classifying different rice varieties. Interestingly, we observed that different features exhibited varying degrees of effectiveness for this classification task. Among the features examined, Green emerged as the most influential and impactful in accurately discerning rice varieties. Remarkably, this feature surpassed well-established vegetation indices, such as the LCI, GNDVI, NDVI, and LAI, in its ability to accurately classify the rice varieties treated in the present study. This finding highlights the significance of this feature and its potential to capture the essential characteristics that differentiate rice varieties.

Our results shed light on the importance of considering a diverse range of features when undertaking variety classification. Although vegetation indices have traditionally been relied upon for this purpose, our results emphasize that alternative features, such as the Green feature, can offer superior performance and contribute significantly to the accurate classification of rice varieties.

The selection of Green reflectance as the most important feature in determining rice variety can be attributed to several factors. First, Green reflectance captures vital information about vegetation health and vigor, as it represents the amount of light reflected by the plants in the relevant wavelength range. This feature is particularly meaningful for rice varieties, as their growth and development rely heavily on chlorophyll content, which affects their overall health and productivity. Higher Green reflectance values indicate healthier and more vigorous vegetation, which is indicative of specific rice varieties.

Second, the Green reflectance feature may possess unique spectral characteristics that are particularly discriminative for differentiating rice varieties. It can capture subtle differences in leaf structure, pigmentation, or physiological traits among the varieties, which are not adequately captured by other vegetation indices such as the LCI, GNDVI, NDVI, or LAI. These indices often focus on specific wavelengths or combinations of wavelengths, whereas Green reflectance provides a more comprehensive representation of the overall greenness of vegetation. Fu et al. [76] suggested that leaf greenness could potentially function as a convenient indicator for identifying genotypes with elevated photosynthetic capabilities.

In addition to its importance as a predictor, the Green reflectance parameter exhibits noteworthy intraclass variability. As mentioned earlier, the variability of a feature within each class provides valuable insight into its discriminative power. In the context of our study, Green reflectance demonstrated a significantly lower standard deviation across all growth stages when compared to other features. Figure 6 visually represents the distribution of Green reflectance and highlights its narrower variability. This reduced variability suggests that Green reflectance possesses distinct and consistent spectral characteristics that enable it to effectively differentiate among different rice varieties. The ability to discern subtle variations in Green reflectance across growth stages makes it a reliable feature for classification. This finding emphasizes the importance of the "green" reflectance as a significant contributor to the classification process.

Additionally, the high contribution of the Green reflectance feature could also be influenced by the specific growth stage of the rice plants during twelve WAP. Different growth stages exhibit varying spectral signatures, and during this stage, the Green

reflectance feature may have stronger discriminatory power in distinguishing rice varieties. It is essential to consider the growth dynamics of rice plants and the corresponding physiological changes that occur at different stages when interpreting the significance of these features. The visualization provided in Figure 8 further supports the importance of the Green reflectance feature, demonstrating its prominent position as an essential predictor. The potential decrease in the AUC of the classification in the absence of this feature highlights its critical role in achieving accurate and reliable classification results for rice varieties.

These findings indicate that different features may be important for classifying different rice varieties. This information can be valuable for designing more effective machine learning algorithms for classifying rice varieties based on remote sensing data. For example, algorithms that weigh certain features more heavily for certain varieties may be able to achieve higher accuracy in classifying these varieties. Overall, this analysis provides important insights into the relationship between different features and classification of different rice varieties, which can inform future research in this area.

This study achieved its highest classification accuracy (0.644) using data from the twelfth WAP, which is significant within the context of this research methodology. As there are no prior studies available for comparison, this outcome is promising and underscores the novelty of our approach, suggesting potential enhancements through further research. Future investigations should explore alternative vegetation indices in conjunction with multispectral bands to improve classification. Deep learning algorithms, renowned for their ability to automatically extract relevant features, hold promise for this task owing to advancements in hardware and software technologies and the abundance of available data. Additionally, evaluating various paddy rice varieties can bolster the model's capabilities and enhance the robustness of the research for reliable analysis of the study area.

## 5. Conclusions and Future Works

In conclusion, this study demonstrated the effectiveness of machine learning algorithms, particularly neural networks, in classifying different paddy rice varieties from the data taken by UAV sensors. The twelfth WAP period was identified as the optimal growth stage for accurate variety classification, considering the significant morphological differences and reduced intraclass variability. The Green reflectance parameter emerged as the most influential predictor, surpassing traditional vegetation indices, owing to its ability to capture essential information about vegetation health and discriminate spectral characteristics for this specific growth stage of rice plants. The combination of leveraging distinctive features and reduced intraclass variability contributed to the improved accuracy of the machine learning algorithms in classifying the rice varieties in this study.

Future research could explore alternative vegetation indices and deep learning algorithms to further enhance the classification outcomes. In the future, evaluating additional paddy rice varieties would strengthen the model's capabilities and expand its applicability.

**Author Contributions:** Conceptualization, A.K.W., L.B.P., A.J. and C.H.; methodology, A.K.W.; validation, A.K.W., L.B.P. and C.H.; formal analysis, A.K.W.; data acquisition, A.K.W., A.A.S. and M.B.R.K.; writing—original draft preparation, A.K.W.; writing—review and editing, L.B.P., A.J., C.H. and H.K; visualization, A.K.W.; supervision, L.B.P., A.J., C.H. and H.K.; project administration, A.K.W., A.J. and C.H. All authors have read and agreed to the published version of the manuscript.

**Funding:** The data collection for this research was funded by the Ministry of Education, Research, and Technology of Indonesia through the Prioritas Riset Nasional (PRN) scheme, based on reference number 2987/E4/AK.04/2021. This work was supported by JSPS RONPAKU (Dissertation Ph.D.) Program.

**Data Availability Statement:** The data used to support the findings of this study may be released upon application to the corresponding author.

**Acknowledgments:** We extend our heartfelt appreciation to Balai Besar Penelitian Padi (Indonesia Center for Rice Research) Subang, West Java for generously providing the necessary research facilities that greatly facilitated this study. We are sincerely grateful for their support and collaboration throughout the study. We express our sincere gratitude to the anonymous reviewers for their invaluable feedback and constructive comments, which have contributed significantly to the improvement of this manuscript.

**Conflicts of Interest:** The authors declare no conflict of interest.

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
