# Peer review of "Machine Learning for Precise Rice Variety Classification in Tropical Environments Using UAV-Based Multispectral Sensing"

_agriengineering, doi:10.3390/agriengineering5040123_

Round 1

Reviewer 1 Report

Comments and Suggestions for Authors

Agriengineering-2646952 investigates the application of machine learning algorithms that utilized multispectral sensor data gathered by a UAV to evaluate different varieties of rice. The paper contents are adequate to the scope of Agriengineering. There are, however, several factors that prevent the publication of this article in its present form.

1.      ‘vertical field of view (FOV) of 62o and 49o’ should be 62°

2.      The units of the variable should be added to the pictures. The font in some of the pictures is too small.

3.      Give reasons for the choice of multispectral bands.

4.      Several of the machine learning methods used in this paper are existing or better-known algorithms, and this paper does not appear to propose new methods or improvements to existing algorithms, which leaves much to be desired in terms of innovation.

5.      There are some problems with the reference formatting of the article that need to be revised.

Comments on the Quality of English Language

None.

Reviewer 2 Report

Comments and Suggestions for Authors

Reviewer report

Title: Machine Learning for Precise Rice Variety Classification in 2 Tropical Environments using UAV-Based Multispectral Sensing

Major correction

·         Line 365-362: The focus of this discussion revolves around the various applications of the NN algorithm. However, it is deemed more appropriate to emphasize similar applications within the discussion. Therefore, the authors are kindly requested to include a more extensive discussion on the classification of different varieties or breeds.

Minor corrections

·         Figure 4 - (Unit 2 and 150 Unit 2), it should be unit 1.

·         Table 1 – please check the font size of the last equation

·         Line 184 – please mention the company and headquarters

·         Line 212 – please give the full form of week

·         Line 204 – the abbreviation was not mentioned before in the text

·         Line 208 – 209: please move this sentence at the start of the section

·         Table 3: in foot note, please give full form the highlighted abbreviations.

·         Figure 7: kindly explain the color code used for the bars.

·         Figure 8: mention that this graph is for 12th week

·         Most of the figure quality is insufficient for publication. Please increase it.

·         Figure 9 and: Axis fonts are too small to read

Comments on the Quality of English Language

The English command is commendable.

Reviewer 3 Report

Comments and Suggestions for Authors

This study effectively utilizes machine learning, particularly neural networks, to identify paddy rice varieties, offering valuable insights into precision agriculture. The choice of the twelfth week after planting as the optimal growth stage enhances classification accuracy. The prominence of the "green" reflectance parameter highlights the significance of considering diverse predictors for variety detection. Future research avenues include exploring alternative indices and deep learning techniques. Overall, this research enhances our understanding of remote sensing applications in agriculture, empowering farmers and researchers to make more informed decisions for crop management, yield optimization, and sustainable agricultural practices.

Reviewer 4 Report

Comments and Suggestions for Authors

See the attached PDF

Comments on the Quality of English Language

There are multiple errors in the English of the article waiting to be corrected. Beware the consistency in your writing.

Reviewer 5 Report

Comments and Suggestions for Authors

First of all, my general evaluation of the article is positive. However, some issues should be improved to meet the quality of the journal.

1.      There are a lot of different techniques described in the “2. Methods” section, which is quite difficult to follow. I suggest adding a flowchart or graphical abstract at the beginning of the “Methods” section where all steps are given.

2.      In “2.2. Experimental Plot Design” it explains how 480 samples were obtained, but there are a lot of details in text that are difficult to follow. I suggest adding a table that will summarize the dataset of 480 samples distributed into columns like date of observations, rice varieties, locations, three different growth stages, 24 experimental plots, and 20 random points, where at the bottom will be shown the total value of 480.

3.      The abstract says that “Classification was performed using k-fold cross-validation ten folds” but I couldn’t see any information on this issue in the text.

4.      The text in lines 253-261 and 267-275 is written twice.

5.      In line 254.  Table 4. was referred but in the paper, there is no Table 4.

6.      Figure 10 and 11. should be moved to the Results section. In Discussion findings should be discussed and how these fit with existing literature. Results are not findings.

7.      In Discussion limitations of the study need to be discussed. Results need to be compared with other studies in the literature. You obtained a CA of 0.644 which is not very high in general but maybe for this particular case, it is an achievement. So, you need to convince the reader by providing results from earlier studies. If the particular case hasn’t been studied yet, you need to highlight that.

8.      In section “2.4 Classification Predictors Selection” the algorithms are listed, but the reason why they were chosen was not given.

Reviewer 6 Report

Comments and Suggestions for Authors

The paper presents a case study of the application of UAV-based Multispectral Sensing for the classification of rice crop varieties. While it extensively describes the methods throughout the text and presents the results robustly, the article does not clearly justify its contribution to the state of the art in machine learning-based classifications. Furthermore, there is a mismatch between the content designated as the discussion, results, and conclusion in the paper. Below, I provide some suggestions that the authors may consider to enhance the article’s structure and clarity:

1 - The introduction inadequately explores knowledge gaps. What unique aspects does this research offer compared to existing work? How does it advance scientific knowledge?

2 - Objectives of the study are presented between lines 67 and 77, with redundancy found in lines 100-108. It is advisable to summarize these objectives only at the end of the introduction.

3 - The paragraphs between lines 78-100 appear disjointed.

4 - Further elaboration on the distinctions between INPARI-32, INPARI-33, and INPARI-43 would enhance the article.

5 - It is essential to provide references to classic literature for all classification methods used.

6 - Lines 199-208 read more like a discussion rather than a presentation of the work.

7 - Between lines 214-220 and 223-235, methodological descriptions are presented, rather than results.

8 - I recommend creating a separate 'Discussion' section within the article, distinct from the 'Results' section, as this distinction has been made in the main body of the text, except for lines 199-208.

9 - Lines 322-328 constitute the article's conclusions.

Round 2

Reviewer 1 Report

Comments and Suggestions for Authors

The comments and questions raised by the reviewers have been addressed. This manuscript can be accepted in its current form.

Comments on the Quality of English Language

None.

Author Response

Dear Reviewer,

We would like to extend our heartfelt gratitude for your diligent review of our manuscript. Furthermore, we deeply appreciate your favorable remarks and valuable recommendations. In our continuous efforts to enhance the manuscript's quality, we have made additional adjustments. You may find the revised version attached.

With sincere regards,

Reviewer 5 Report

Comments and Suggestions for Authors

All changes were made as recommended. Thank you.

Author Response

(The authors gave the same response as above.)

Reviewer 6 Report

Comments and Suggestions for Authors

The authors addressed all the queries raised and made important changes to the paper's structure.

Author Response

(The authors gave the same response as above.)
